# Morphological Variation and New Description of the Subcutaneous Gland of *Sepiella inermis* (Van Hasselt, 1835) in Thai Waters

Sonthaya Phuynoi [1,2,3], Charuay Sukhsangchan [3], Ran Xu [4] and Xiaodong Zheng [1,2,*]

[1] Key Laboratory of Mariculture, Ministry of Education, Ocean University of China, Qingdao 266003, China; sonthaya.p59@outlook.com
[2] Institute of Evolution & Marine Biodiversity (IEMB), Ocean University of China, Qingdao 266003, China
[3] Department of Marine Science, Faculty of Fisheries, Kasetsart University, Bangkok 10900, Thailand; charuay44@hotmail.com
[4] College of Animal Science and Technology, Anhui Agricultural University, Hefei 230036, China; ranxu4@ahau.edu.cn
[*] Correspondence: xdzheng@ouc.edu.cn; Tel.: +86-532-82032873

**Abstract:** The external morphology and morphological variations of *Sepiella inermis* vary across regions, necessitating investigation. However, the histological information on the subcutaneous gland has been insufficient to describe it. In this study, specimens were systematically collected and characterized from the Gulf of Thailand. Regarding external morphology, female cuttlebones exhibit greater width and more pronounced curves compared to males, while males feature 17–19 white dots along the fin margins. The presence of the subcutaneous gland was discerned during the embryonic stage at stage 19. A histological study of the subcutaneous gland illustrated the structure and development of the gland in both embryonic and adult stages, with four layers of membranes covering the gland. In the adult stage, trabeculae are dispersed throughout the gland, whereas in the embryonic stage, they form four distinct lines. The morphometric analysis revealed significant differences between males and females ($p < 0.05$) and morphological variations among the seven locality groups within the sexes were observed ($p < 0.05$) in Chonburi Province. According to the discriminant analysis results, there were significant differences ($p < 0.05$) between the groups in Surat Thani Province. Examining the length–weight relationship between dorsal mantle length and body weight showed significant differences between the sexes, indicating an allometric growth.

**Keywords:** spineless cuttlefish; morphology; histology; gland pore; the Gulf of Thailand

## 1. Introduction

Cuttlefish species are currently recognized as one of the most significant fisheries resources, prized for their exceptional nutritional content and high protein levels. Among them, the genus *Sepiella*, belonging to the family Sepiidae, is widely distributed in coastal regions of tropical and sub-tropical zones, including South China sea. In Thai waters, *Sepiella inermis*, commonly known as the spineless cuttlefish, can be found in both the Andaman Sea and the Gulf of Thailand, situated within the Indo-Pacific Ocean. This species holds considerable commercial value in Thailand, often being captured using trawl nets.

The distinctive external morphology of *S. inermis* is characterized by the presence of a cuttlebone, the absence of a spine, and the posterior gland. Some diagnostic features were described in the reports by Jereb and Roper, and Nateewathana [1,2], including certain features such as mantle shape, fins, arms, male hectocotylized arm, and cuttlebone; however, *Sepiella inermis* lacks specific details regarding the shape of the radula, the beak, and certain aspects of the cuttlebone. Consequently, the present study provides these details completely for this species. The two ink glands are present in a single individual. The development of the ink sac spans from embryogenesis to the adult stage [3]. The process

of ink release from the sac begins with stimulation originating from the latero-ventral palliovisceral lobe located within the posterior subesophageal mass of the central mantle cavity. Subsequently, the ink is expelled from the lumen to the exterior [4]. Another ink gland is situated ventrally at the posterior end of the mantle, referred to as a gland pore, pigment gland, or glandular pore in previous literature [5–8]. Despite the considerable volume of research conducted on this species, there is still a notable lack of information regarding the structure and function of the subcutaneous gland [9,10].

Morphological variation encompasses the morphometric differences observed within a group of samples [11], particularly among males and females. This assessment depends on several factors, such as species, environmental variations, and regions, and extends to population structure. For instance, a study conducted in Indonesia [12] identified significant morphometric variations and discriminant analysis among three cephalopod species (*Sepioteuthis lessoniana*, *Sepia officinalis*, and *Uroteuthis* sp.). Notably, *S. officinalis* and *S. lessoniana* exhibit morphometric similarities despite belonging to different orders. In this investigation of morphometric measurements, dorsal mantle length (DML) was widely accepted as a standard [13]. Additionally, total body weight serves as a typical metric for morphological and growth rate evaluations. The analysis of morphological variation is crucial for estimating growth rates in both males and females. Oceanographic variables have significantly contributed to the migrations, distributions, populations, and density of each cephalopod species within specific geographical areas [14].

In Thai waters, several key factors contribute to oceanographic and environmental variables, including diverse biological features in different locations and variations in nutrient compositions. This study uses length–weight relationship analysis to evaluate the growth rates of *S. inermis* in this region [15–17]. For instance, *Sepia officinalis* and *Sepia elegans* in the English Channel, Spain, Ría de Vigo, and Gulf of Cadiz exhibited allometric growth [18], even though they are the same species in different areas with similar growth rates. Therefore, habitat differentiation has been closely linked to the observed variations in morphology, growth rates, and cephalopod population structures in each respective area [19–22]. In Thailand, there is currently a scarcity of data regarding the growth of this species.

The objectives of this article encompass a comprehensive analysis of the external morphological traits, morphological variations, and length–weight relationship of *S. inermis* in Thai waters. Furthermore, it is crucial to examine the histological structure of the subcutaneous gland in this species. This research is of significant importance, as the subcutaneous gland stands out as one of the defining characteristics of this species and represents the broader genus *Sepiella*. By delving into these aspects, this study aims to deepen our understanding of *S. inermis* and contribute valuable insights to the knowledge base about this species.

## 2. Materials and Methods

### 2.1. Sample Collection

A total of 396 fresh specimens of *S. inermis* were collected from the waters of Thailand, comprising 203 males and 193 females. The study area was divided into seven coastal locations. Within the Gulf of Thailand, specimens were gathered from six stations, with 30 (16 females, 14 males) specimens from Trat (T), 61 (26 females, 35 males) from Chonburi (CB), 80 (4 females, 40 males) from Samut Songkhram (SSM), 75 (25 females, 50 males) from Phetchaburi (PT), 30 (20 females, 10 males) from Surat Thani (SNI), and 45 (18 females, 27 males) from Songkhla Province (SK). Additionally, one station in Ranong Province (RN) contributed 75 (48 females, 27 males) specimens from the Andaman Sea in Tables A1 and A2 and Figure 1. Eggs of *S. inermis* were collected from natural sources, specifically fishing gear, and were subsequently hatched into paralarvae in tanks for the histological study of the subcutaneous gland.

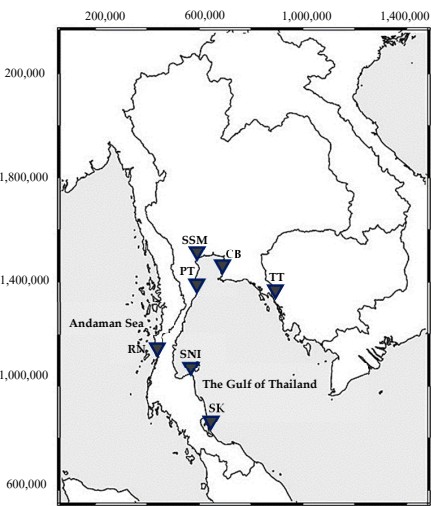

**Figure 1.** Sampling study area and positions where *Sepiella inermis* were collected for morphological characteristic analysis in the waters of Thailand.

## 2.2. Analysis of Morphological Indices

Quantitative variables of cuttlefish morphology include the following 20 morphological characteristics [6]: body weight (BW), dorsal mantle length (DML), dorsal mantle width (DMW), total length (TL), cuttlebone length (CL), cuttlebone width (CW) (Figure 2c), cuttlebone weight (CWt), head length (HL), head width (HW), eye diameter (ED), funnel length (FL), funnel width (FW), length of the left arms 1 (LI), length of the left arms 2 (LII), length of the left arms 3 (LIII), length of the left arms 4 (LIV), length of the right arms 1 (RI), length of the right arms 2 (RII), length of the right arms 3 (RIII), and length of the right arm.

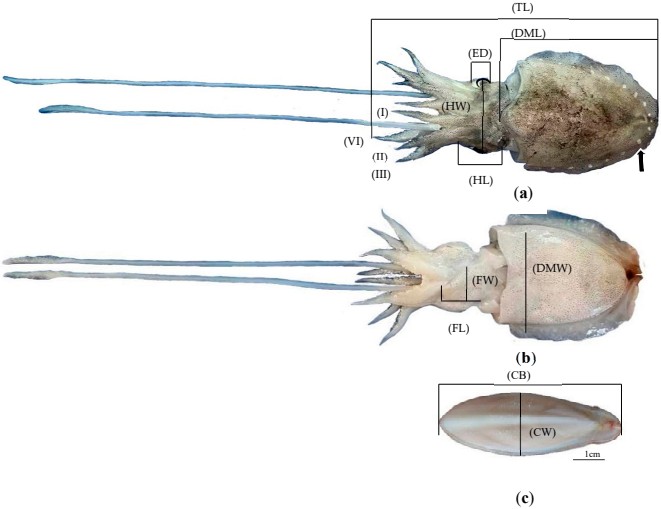

**Figure 2.** Measurement method for assessing the external morphological characteristics of *Sepiella inermis*: (**a**) measurement of the dorsal mantle with a specific focus on the presence of white spots in males, (**b**) measurement of the ventral mantle, and (**c**) measurement of the dorsal cuttlebone. It should be noted that the abbreviations used in the measurements are as follows: TL represents total length, DMW denotes dorsal mantle width, CB represents cuttlebone, CW denotes cuttlebone width, DML refers to dorsal mantle length, ED stands for eye dimension, FW represents funnel width, FL denotes funnel length, HL represents head length, and HW refers to head width. Additionally, measurements include the length of the left arms (I, II, III, and VI) and the length of the right arms (I, II, III, and VI) [6]. (The black arrow showed White spots in males).

Measurements of these morphological features were taken for each sample to the nearest millimeter, following the method described by [6]. The weight of the cuttlefish

was recorded to the nearest milligram, representing the body weight (BW) and cuttlebone weight (CWt). To analyze the statistical significance of the external morphological traits, one-way analysis of variance (ANOVA) and *t*-test samples were conducted using SPSS version 27 (V 27-c14f0926747987220be0) software, ensuring accurate and robust statistical analysis of the data. To facilitate a more comprehensive analysis of the data and facilitate the visualization of group differences, principal component analysis (PCA) was used to explore the differentiation in external morphological indices based on sexes and study areas used by SPSS version 27 and Origin software. This approach allows for the creation of graphs that enhance the interpretation of the results. In addition, discrimination analysis was performed to classify and identify different groups based on sex and study areas. Fisher's criterion and cluster analysis were employed in this process, utilizing Euclidean distance by SPSS version 27. The accuracy of the discrimination analysis was assessed using the following formula:

$$P_1 = \text{the number of individual samples in each study area/the total number of the samples} \times 100\%;$$

$$P_2 = \text{the number of individual samples in each study area/the number of samples of this species in the group} \times 100\%$$

$$\text{Comprehensive discrimination rate} = \sum_{i}^{k} = 1A_i / \sum_{i}^{k} = 1B_i$$

where $A_i$ and $B_i$ represent the correct number of individual samples in the groups and the number of individual samples in this species actually discriminated, respectively, and *k* is the number of groups Table A2.

The data from the length–weight relationships were analyzed with SPSS version 27 by gender using the equation:

$$W = aL^b$$

where W = weight, L = length, a = scaling coefficient for weight at sample length, and b = shape parameter for the body form of the sample.

### 2.3. External and Internal Structure Characteristics

2.3.1. Scanning Electron Microscopy (SEM)

For the study of the external structure of the freshly hatched paralarvae stage obtained from nature, scanning electron microscopy (SEM) was employed. The specimens underwent a dehydration process utilizing a series of ethanol concentrations: 75%, 80%, 85%, 90%, 95%, and 100%, with each concentration lasting for 15 min. This dehydration process was conducted in triplicate (three replications) to ensure accuracy and consistency. The dehydrated samples were then covered with paper and placed in a critical point drier (CPD) machine for 90 min. After the samples were thoroughly dried, they were mounted on a metal stub using a sticky carbon disc and kept in a desiccator for 72 h to ensure optimal preservation. To enhance the imaging quality, the dried samples were coated with white gold platinum using the quorum Q150R Thin-Film Coater (quorum q150 res) for 60 min before the examination. The examination of the prepared samples was carried out using a scanning electron microscope (SEM-Hitachi SU8020). The methodology used in this study follows that of [23].

2.3.2. Histological Techniques

Histological techniques were employed to examine the histological structure of the subcutaneous gland of *S. inermis* during the paralarvae stage (3 days after hatching, with a dorsal mantle length of 2.45 ± 0.12 cm). The gland was carefully dissected and immediately fixed in 10% buffered formalin. To prepare the tissue for histological analysis, a dehydration process was conducted using a series of ethyl alcohol concentrations, gradually increasing from low to high. This was followed by a clearing procedure involving the use of a xylene solution. Subsequently, the tissue was infiltrated with paraffin wax at 56–60 °C for 24 h

until it was fully embedded. To obtain thin sections suitable for microscopic examination, longitudinal and transverse histological sections with a thickness of 5 μm were prepared using the RMC Boeckeler MR3 motorized rotary microtome. The sections were then stained using a series of ethanol, hematoxylin, and eosin (H&E) to enhance contrast and visualization. Finally, the histological structure of the subcutaneous gland was examined using a light compound microscope (Olympus CX43), allowing for detailed observations and analysis.

## 3. Results

### 3.1. Morphological Characteristics of Sepiella inermis

Description: The dorsal mantle length (DML) size of *S. inermis* varies between 32 and 92 mm (59.07 ± 10.51 mm). The measurements of the remaining morphology are represented as a percentage of the DML: dorsal mantle width (77.29% DML); body weight (39.37 ± 18.40 mg); dorsal total length (107.83 ± 21.35 mm, 54.78% TL); cuttlebone length (94.49% DML); cuttlebone width (39.15% DML); cuttlebone weight (3.42% BW); head width (50.05% DML); head length (33.52% DML); eyes small (29.59% HL); the funnel moderate long shape (52.10% DML); funnel width (18.01% DML); the arms of males are slightly longer than females, about 14.36 mm, with the length of the left arms 1 (41.08% DML), the left arms 2 (40.62% DML), the left arms 3 (44.0 0% DML), the left arms 4 (55.40% DML), and the right arms 1 (41.17% DML), the right arms 2 (41.32% of DML), the right arms 3 (44.32% DML), and the right arms 4 (54.74% DML). The cuttlebone exhibits a broad oval shape and lacks spines. The outer cone resembles a semicircle, and the inner cone is thick and rounded. The ventral surface appears convex, while the dorsal surface appears rough and convex. The outer cone of female cuttlebones is broader and more curved compared to males, as depicted in Figure 3a,b. The arm formula adheres to the sequence of IV > III > II > I. Notably, the outer margins of the hectocotylized arm on the left ventral arm IV exhibit a textured appearance characterized by transverse folds, as shown in Figure 4a. Furthermore, differentiation between male and female specimens can be achieved by observing the presence of 17–19 distinctive white spots along the fin margins. The upper beak (Figure 4b) is a sharp, long, and curved rostrum with a prominent large hood; its tip is black. The lateral wall displays a gentle curve, ranging from white to brown, and includes a sizable plate. The lower beak has a less curved and blunt rostrum tip. The lower hood is shorter and narrower. The front of the inner jaw angle is distinctly black, and the two wings are elongated, exhibiting a slight brown hue. The radula comprises seven transverse rows of teeth. The rachidian tooth (RA) base width is 163.04 and 182.30 μm in length, followed by the first lateral tooth (L1) 189.13 μm wide and 195.65 μm in length, the second lateral tooth (L2) 166.30 μm wide and 234.78 μm in length, and the marginal tooth (MT) 146.93 μm wide and 404.08 μm in length. The rachidian tooth is characterized by a wide and symmetrical base. The marginal tooth is elongated and takes on a cusp-shaped form, being the largest tooth in the set. Notably, the basal structure of the marginal tooth forms a wide and smooth plate as shown in Figure 5a–e.

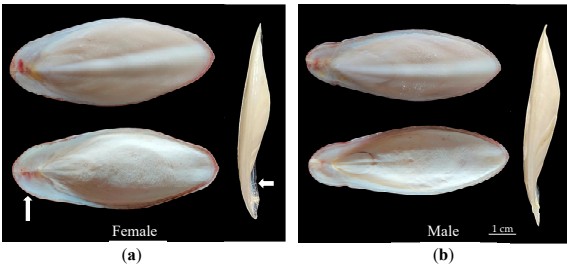

**Figure 3.** Differentiation of cuttlebone between females and males of *Sepiella inermis*: (**a**) The outer cone of female cuttlebones is broader and more curved males (White arrow indicates ventral lateral view and curved in cuttlebones). (**b**) The outer cone of the male cuttlebone is narrower and less curved.

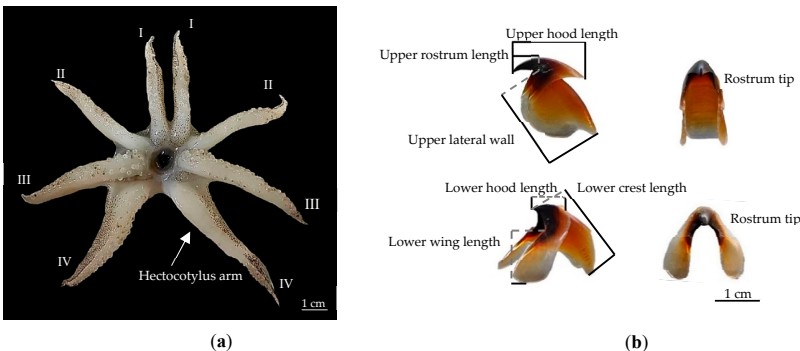

(a)

(b)

**Figure 4.** External morphological characteristics of *Sepiella inermis*. (**a**) Arms form as follows: I, II, III, IV, and hectocotylus arm, which is the left ventral in males (white arrow indicates hectocotylus arm). (**b**) The details of upper beak and lower beak.

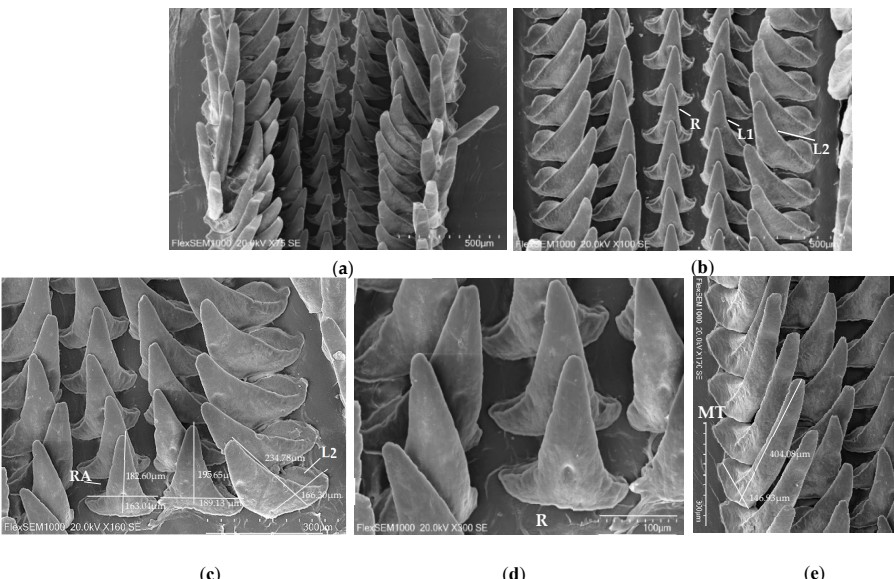

**Figure 5.** Radula features of *Sepiella inermis*: (**a**–**c**) radula showing a transverse row of RA; RA—rachidian tooth, L1—first lateral, L2—second laterals, MT—the marginal tooth; (**d**) showing the shape of rachidian tooth; and (**e**) showing the marginal teeth feature are long and cusp-like.

### 3.2. The Subcutaneous Gland's Structure

The structure of the subcutaneous gland can be categorized into two parts: its external and internal morphological development from embryonic to adult stages. Remarkably, during the embryonic stage, the external morphological structure of the subcutaneous gland was observed. Further details outlining this developmental process can be found in Table 1 and Figure 6.

The external characteristic features of the gland became more discernible through an examination using a scanning electron microscope (SEM). The preservation of the samples in this process enhances the observation of a more detailed external mantle morphology of the gland. Notably, four lobes and grooves are symmetrically positioned between the under fins, as depicted in Figure 7a,b.

**Table 1.** External morphological feature of the subcutaneous gland in embryonic stages of *Sepiella inermis* (observed in stage 19).

| Stages | Description of the Formation of the Subcutaneous Gland |
|---|---|
| 19 | The subcutaneous gland form has a small knob at the end of the mantle. |
| 20 | The gland is clearly divided into two lobes, light brown, slightly convex, and resembles a cuttlebone. |
| 21 | Two lobes are more convex than in stage 20 and are light orange, and an ink sac is absent. |
| 22 | Two lobes that have grown increasingly convex, starting to take on a darker orange color, and developing an ink sac form. |
| 23 | The gland's two lobes have grown more convex, and four lines form within the internal structure. |
| 24 | The gland has two larger, dark orange lobes, a longer internal lines structure, and a clear permanent symmetry. Forming the first four lobes. The ink sac and ink duct are visible. |
| 25 | The ventral side of the gland displays four lobes. The gland has changed to dark orange instead of orange in color. Fins, ink duct, and ink sac are formed completely. |
| 26 or Hatching stage | The subcutaneous gland's structure is clearly visible and the four-line internal structure is located within four lobes on the ventral side. Melanin pigment is contained in the gland. |

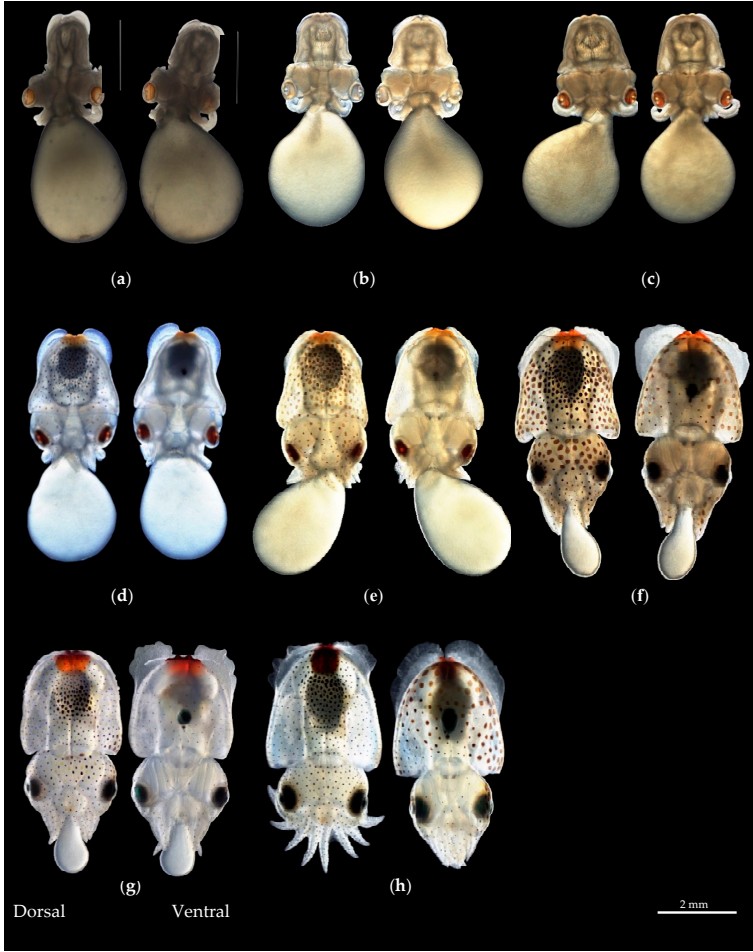

**Figure 6.** External shape and morphological characteristics of the subcutaneous gland of *Sepiella inermis* during different embryonic stages: (**a**) In stage 19, the gland appears as a small knob with no color. (**b**) In stage 20, two lobes begin to form, accompanied by a slight brown coloration. (**c**) By stage 21, the lobes continue to grow and exhibit a light orange hue. (**d**) During stage 22, the lobes become increasingly convex and dark orange, and an ink sac starts to form. (**e**) In stage 23, the lobes reveal four distinct lines and adopt an orange coloration. (**f**) Stage 24 is characterized by a longer internal line structure, the presence of four lobes, and the completion of the ink sac and duct. (**g**) By stage 25, four lobes are clearly visible. (**h**) In the hatching stage, four lobes can be observed on the ventral side of the gland, which contains melanin.

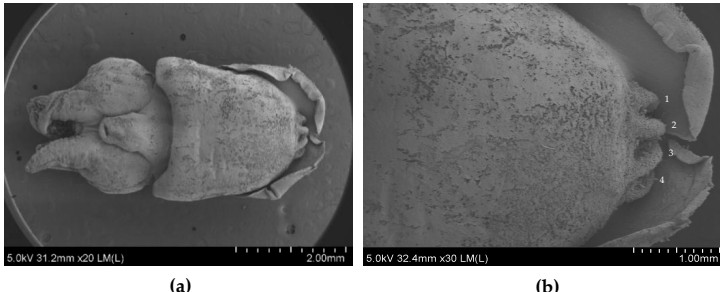

**Figure 7.** SEM images of the external morphological characteristics of *Sepiella inermis* embryonic stages: (**a**) the hatching stage, and (**b**) the subcutaneous gland comprises four lobes, with lobes 2 and 3 forming a central pair larger than lobes 1 and 4, located on the ventral side of the mantle.

### 3.2.1. Internal Structure Subcutaneous Gland at Hatching Stage

The internal structure of the subcutaneous gland was examined during the paralarvae stage. The gland exhibits a circular shape in the transverse section in Figure 8a. A thin mantle layer covers the entire dorsal mantle, including the cuttlebone portion. The subcutaneous gland's structure is complex and described as follows:

- The wall of the subcutaneous sac consists of four layers: (1) The epidermis is the outermost layer protected by an external membrane. (2)The middle layer comprises connective tissue and chromatophores pigments. (3) The muscularis is a thick layer of muscle fibers. (4) The sac's interior is coated by mucosa and cuboidal cells. Melanin pigment is spread along the margin of the wall, as depicted in Figure 8b.
- Within the sac of the wall is a large cavity containing a high quantity of perforate lamellae resembling nerve nets from connective tissue. The basal subcutaneous gland is a production source melanin glandule, comprising four different trabeculae branches of connective tissue in the cavity Figure 8c. These branches consist of two lengthy ones extending from the center basal gland to the lumen, as shown in Figure 8d. All branches from the basal subcutaneous gland are interconnected. Each branch of trabeculae contains melanin granule, as referred to in Figure 8f.
- The gland pore is controlled by a muscle that regulates ink release. The mantle musculature and ventral fins regulate the release and transmission of liquid ink refer to Figure 8e.

### 3.2.2. Internal Structure of Subcutaneous Gland at Adult Stage

During the adult stage, the subcutaneous gland undergoes a change, adopting a pear-shaped structure within the sac's lumen. The posterior spot of the gland is notably brown-black and the measured length was 5–12 mm (8.67 ± 2.46 mm) and the width ranged from 2.2 to 5.7 mm (4.05 ± 1.20 mm) (Figure 9a,c).

The adult stage's internal structure is composed of four layers of the muscular region: (1) the mantle layer of outer muscle; (2) connective tissue; (3) circular muscle fibers that are longitudinal and extend throughout the entire gland; and (4) the inner wall is constructed of a membrane border that surrounds the sac and is connected with trabeculae lines through the gland. The lumen of the subcutaneous gland adopts the form of an elongated tube and is surrounded by muscle tissue. This anatomical arrangement is depicted in Figure 9a,c. Melanin glandules penetrate the membrane, dispersed around the lateral surface of the trabeculae. Consequently, the trabeculae connective tissue related to the inner wall became the primary source of melanin glandule production in the adult stage (Figure 9d,e). Figure 9f shows the distribution of melanin within the trabeculae connective tissue, which causes black pigmentation within the gland.

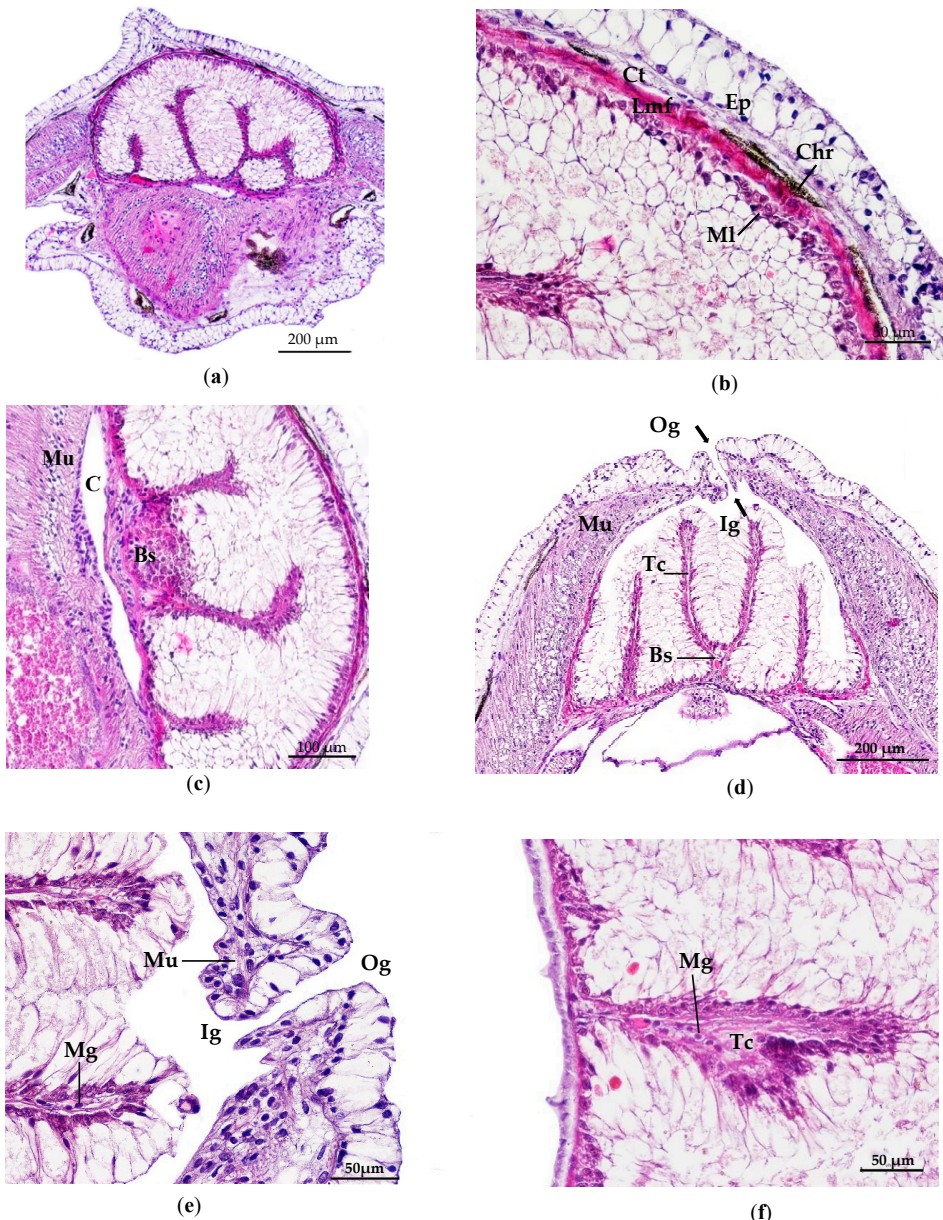

**Figure 8.** Histological structure of subcutaneous gland in the hatching stage of *Sepiella inermis*: (**a**) transverse section and the shape of the subcutaneous gland is circular, (**b**) the wall membrane are four layers on transversal section, (**c**) transverse section showing the large basal subcutaneous gland (Bs) in central of cavity, (**d**) four-line branches is connected with the basal, (**e**) horizontal section; the ink in the subcutaneous gland is released through this lumen, and (**f**) melanin granule cell. Abbreviations: Bs—basal region of subcutaneous gland, C—cuttlebone, Chr—chromatophore, Ct—connective tissue, Ep—epidermis, Ig—internal gland pore, Lmf—longitudinal muscle fiber, Ml—mucosa lines, Mg—melanin glandular cells, Mu—muscular layer, Og—outer gland pore, Tc—trabeculae of connective tissue. Stained with hematoxylin and eosin.

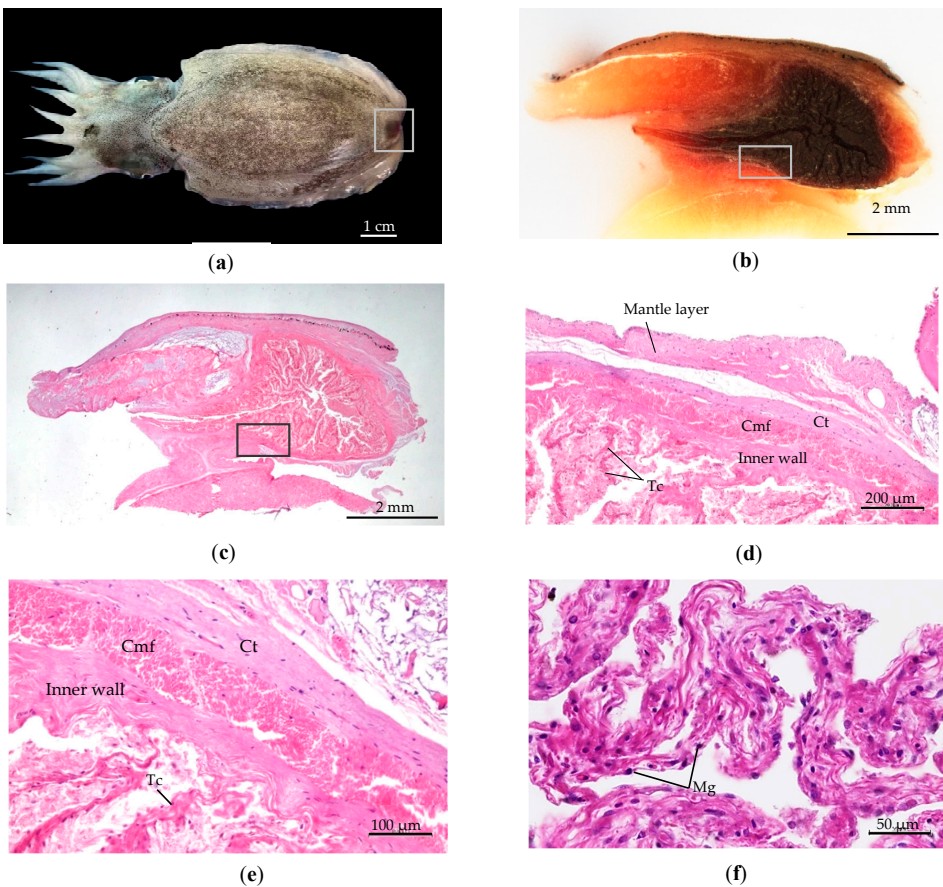

**Figure 9.** Subcutaneous gland in the adult stage of *Sepiella inermis*: (**a**) showing that the structure of the subcutaneous gland is blocked by paraffin and melanin pigment in a cavity (the gray rectangle presents the subcutaneous gland), (**b**) position of subcutaneous gland in adult stage, (**c**) transverses section whole histological structure of subcutaneous gland in adult (the rectangle in **b** and **c** indicates the position of the cross-section in the subcutaneous gland), (**d**,**e**) histological structure layer membranes, and (**f**) glandular tissue comprise pigments distributed into trabeculae connective tissue. Abbreviations: Ct—connective tissue, Cmf—circular muscle fibers, Mg—melanin glandular cells, Tc—trabeculae of connective tissue. Stained with hematoxylin and eosin.

### 3.3. Morphological Variations Analysis and Length–Weight Relationship

#### 3.3.1. Morphological Variations of *S. inermis*

Detailed information on 20 morphological traits of *S. inermis*, including size and gender, is listed in Tables A1 and A2. Overall, the size and gender analysis indicate that the 203 observed males exhibited body weights ranging from 11 to 82 g, and dorsal mantle lengths (DML) ranging from 35 to 79 mm (56.91 ± 9.16 mm). Conversely, the 193 females displayed body weights ranging from 8.74 to 133 g, with DML ranging from 32 to 92 mm (61.34 ± 11.35 mm). Principal component analysis (PCA) was conducted to assess morphological variations among different localities and sexes. The analysis revealed significant distinctions ($p < 0.05$) in morphological traits within both sexes, leading to the classification of populations into three distinct groups. Group 1 comprises T, SK, and SNI stations, while Group 2 consists of SSM, PT, and RN stations. Group 3 is the CB station, as illustrated in Figures 10a and 11, and detailed in Table A2.

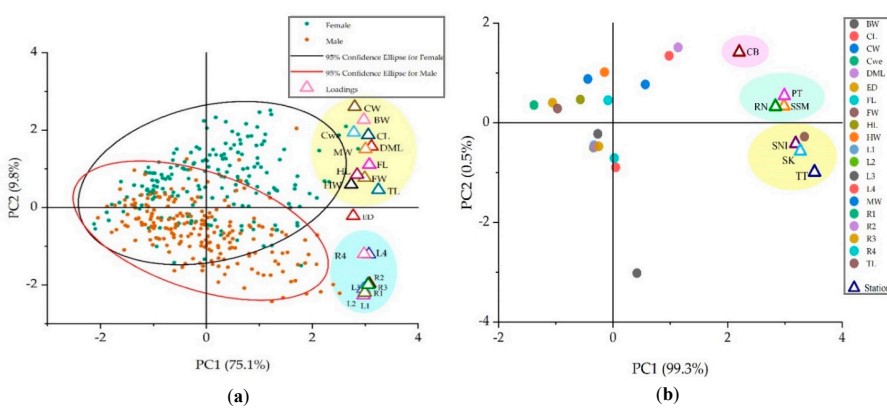

(a)  (b)

**Figure 10.** Principal component analysis. (**a**) Distribution of morphological characteristic proportions in both male and female specimens of *Sepiella inermis* within each locality. Distinct patterns and clusters can be observed, indicating differentiation in external morphology among the specimens. Thin circles: differentiation traits; solid circles: samples categorized by males and females at a 95% confidence interval. Notably, the analysis successfully discerns sexual differentiation between females and males. (**b**) Relationship be tween the morphological indices of both sexes and the seven populations. The samples are classified into three distinct groups: (1) TT, SK, and SNI stations, represented by orange circles; (2) SSM, PT, and RN stations, indicated by light blue circles; and (3) CB station, denoted by solid circles in purple. Solid circles: samples categorized by study areas and color of spots are the morphological traits at a 95% confidence interval.

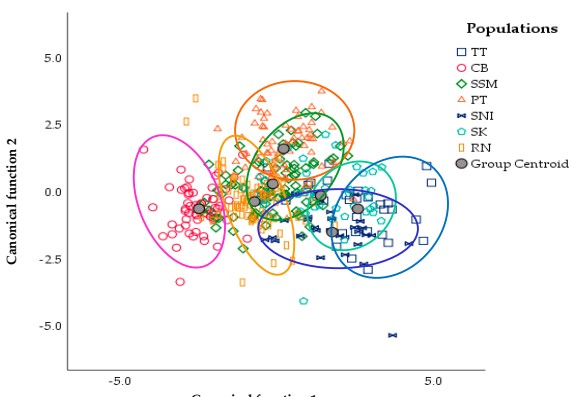

**Figure 11.** Canonical discriminant scatter plots of *S. inermis* populations in seven geographical regions. Thin circles indicated differentiation of study areas at a 95% confidence interval.

The discrimination analysis of female and male cephalopods yielded significant findings [24–26]. The percentage of discriminant accuracy for seven groups of females ranged from 55% to 100%. For males, the discriminant accuracy ranged from 65% to 100%, as shown in Tables 2 and 3. Notably, the station SNI exhibited the highest differentiation rate within Group 1, indicating a distinct difference from other groups. Conversely, SK demonstrated the lowest discrimination accuracy among females, while SSM had the lowest accuracy among males, as detailed in Tables 2 and 3. Cluster analysis revealed statistically significant differences ($p < 0.05$) between females and males across the seven localities, as illustrated in Figure 12. The analysis showed that the seven localities or population groups could be categorized into three female clustering groups: Group 1 consists of PT, SK, SNI, and SSM; Group 2 is T; and Group 3 comprises CB and RN. In terms of male clustering groups, Group 1 consists of SSM, PT, SNI, and CB; Group 2 is SK; and Group 3 comprises T and RN. Moreover, diagram analysis yielded similar results for both sexes at the SNI and RN stations.

**Table 2.** Discrimination result of female *Sepiella inermis* populations in Thailand. Abbreviated names of the localities: T—Trat, CB—Chonburi, SSM—Samut Songkhram, PT—Phetchaburi, SNI—Surat Thani, SK—Songkhla, RN—Ranong Province.

| Stations | Number of Samples | Predict Classification of Female | | | | | | | Discrimination Accuracy% |
|---|---|---|---|---|---|---|---|---|---|
| | | T | CB | SSM | PT | SNI | SK | RN | |
| T | 16 | 12 | 0 | 2 | 0 | 2 | 0 | 0 | 75.0 |
| CB | 26 | 0 | 22 | 0 | 0 | 0 | 0 | 4 | 84.6 |
| SSM | 40 | 3 | 5 | 22 | 1 | 0 | 0 | 9 | 55.0 |
| PT | 25 | 0 | 0 | 1 | 22 | 1 | 1 | 0 | 88.0 |
| SNI | 20 | 0 | 0 | 0 | 0 | 20 | 0 | 0 | 100.0 |
| SK | 18 | 1 | 0 | 3 | 1 | 1 | 10 | 2 | 55.6 |
| RN | 48 | 0 | 4 | 4 | 0 | 0 | 1 | 39 | 81.3 |
| Comprehensive Discrimination Rate: 71.5% | | | | | | | | | |

**Table 3.** Discrimination result of male *Sepiella inermis* populations in Thailand. Abbreviated names of the localities: T—Trat, CB—Chonburi, SSM—Samut Songkhram, PT—Phetchaburi, SNI—Surat Thani, SK—Songkhla, RN—Ranong Province.

| Stations | Number of Samples | Predict Classification of Male | | | | | | | Discrimination Accuracy% |
|---|---|---|---|---|---|---|---|---|---|
| | | T | CB | SSM | PT | SNI | SK | RN | |
| T | 14 | 13 | 0 | 1 | 0 | 0 | 0 | 0 | 92.9 |
| CB | 35 | 0 | 33 | 2 | 0 | 0 | 0 | 0 | 94.3 |
| SSM | 40 | 0 | 5 | 26 | 4 | 0 | 1 | 4 | 65 |
| PT | 50 | 0 | 0 | 4 | 39 | 2 | 1 | 4 | 78 |
| SNI | 10 | 0 | 0 | 0 | 0 | 10 | 0 | 0 | 100 |
| SK | 27 | 0 | 0 | 2 | 1 | 0 | 23 | 1 | 85.2 |
| RN | 27 | 0 | 2 | 0 | 2 | 1 | 0 | 22 | 81.5 |
| Comprehensive Discrimination Rate: 81.8% | | | | | | | | | |

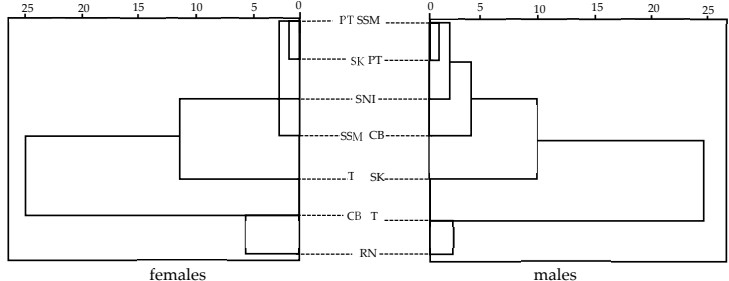

**Figure 12.** Diagram of cluster analysis of seven populations of *S. inermis*. Abbreviated names of the localities: T—Trat, CB—Chonburi, SSM—Samut Songkhram, PT—Phetchaburi, SNI—Surat Thani, SK—Songkhla, RN—Ranong Province.

Additionally, Figure 10a illustrates that the distribution of morphological characteristic proportions in females is broader than in males. The relationship between dorsal mantle length and the sexuality of *S. inermis* was explored, revealing that females typically exhibit larger than dorsal mantle lengths in Table A1, ranging between 40 and 49 mm ($t = 6.257$, $p < 0.05$), compared to males, whose lengths range from 30 to 39 mm ($t = 18.500$, $p < 0.05$).

3.3.2. Length–Weight Relationship

In this study, the length–weight relationship of both sexes was investigated. Females exhibited a significant variance in length ($t = 30.900$, $p < 0.05$) and were heavier ($t = 75.096$, $p < 0.05$) than males. The morphometric parameters between dorsal mantle length (DML) and body weight (BW) showed a high correlation and were significantly different in males

and females: $r = 0.917$ ($p < 0.05$). When both sexes were combined, the correlation ($r = 0.906$) remained highly significant ($p < 0.05$). Additionally, the analysis of covariance (ANCOVA) yielded a significant result ($F = 32.367$, $p < 0.05$). These variations in length and weight represent clear distinctions between the sexes. The mode of growth was reflected in the *b* value of males ($b = 2.1746$) and females ($b = 2.3837$), indicating allometric growth, and both sexes combined data illustrated allometric growth ($b = 2.3616$) as well in Figure 13 and Table 4. Overall, the results indicate that in both sexes, total body weight grows at a slower rate than dorsal mantle length throughout the ontogeny of *S. inermis*.

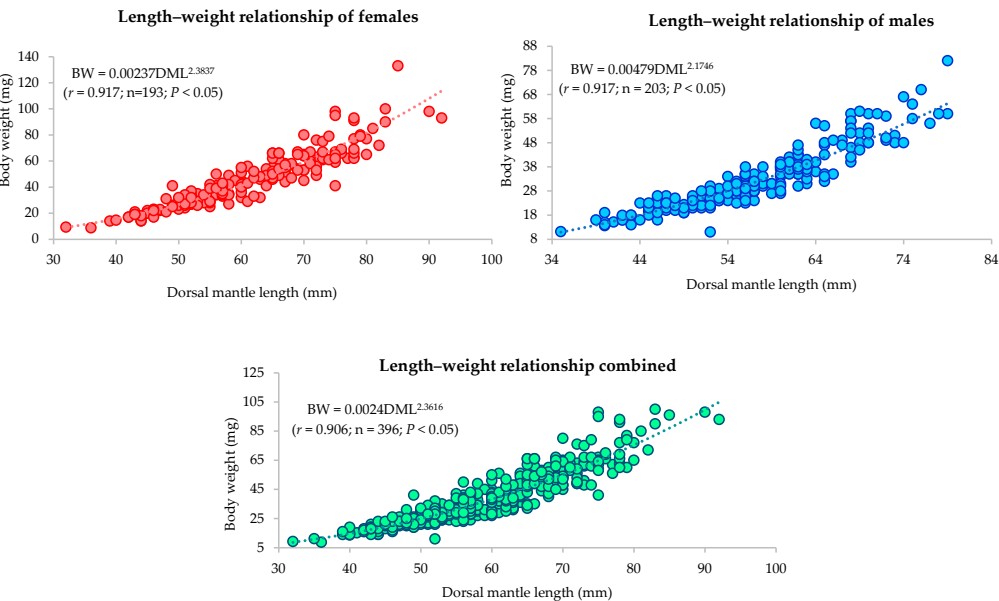

**Figure 13.** Length–weight relationships for females, males, and both sexes combined of the spineless cuttlefish *S. inermis* from Thai waters.

**Table 4.** Length–weight relationship and relative growth of *S. inermis*.

| Sex/ Stations | Length–Weight Relationship Parameters | a | b | r | Growth Relationship |
|---|---|---|---|---|---|
| Female | BW = 0.00237 × DML$^{2.3837}$ | 0.00237 | 2.3837 | 0.917 | Allometric |
| Male | BW = 0.00479 × DML$^{2.1746}$ | 0.00479 | 2.1746 | 0.917 | Allometric |
| Combined | BW = 0.00242 × DML$^{2.3616}$ | 0.00242 | 2.3616 | 0.906 | Allometric |
| TT | BW = 0.00112 × DML$^{2.5507}$ | 0.00112 | 2.5507 | 0.938 | Allometric |
| CB | BW = 0.00786 × DML$^{2.0382}$ | 0.00786 | 2.0382 | 0.880 | Allometric |
| SSM | BW = 0.00462 × DML$^{2.1990}$ | 0.00462 | 2.1990 | 0.904 | Allometric |
| PT | BW = 0.00134 × L$^{2.4789}$ | 0.00134 | 2.4789 | 0.909 | Allometric |
| SNI | BW = 0.000174 × L$^{3.0272}$ | 0.000174 | 3.0272 | 0.914 | Isometric |
| SK | BW = 0.00171 × DML$^{2.4513}$ | 0.00171 | 2.4513 | 0.822 | Allometric |
| RN | BW = 0.00121 × DML$^{2.5622}$ | 0.00121 | 2.5622 | 0.926 | Allometric |

In addition, a comprehensive analysis of the length–weight relationship across all stations revealed a significant difference ($p < 0.05$). The relative growth rate in six stations RN, T, PT, SK, and CB showed allometry patterns, indicating that samples in these stations have a bodyweight growth rate slower than the length of the dorsal mantle. However, only the SNI station exhibited an isometric growth rate ($b = 3.0272$), implying a direct positive proportion between weight and length in Table 4.

## 4. Discussion

### 4.1. Morphological Characteristics of S. inermis

This study has presented a comprehensive description of the external morphological characteristics of *S. inermis*, emphasizing the presence of sexual differentiation. A distinct

feature observed in males is the presence of white spots around the margin of fins, typically numbering between 17 and 19. Previous research by [15] indicates that white spots appear on dorsal mantle length at approximately 35 mm in males. Additionally, the forms of female cuttlebones are wider and exhibit dents compared to males in the ventral lateral view. Cuttlebone investigation was unable to identify sexual differentiation because of variations in growth rate and geographic locations [27]. The upper and lower beaks of *S. inermis* resemble those of *Sepiella japonica*, featuring a sharp and black color at the rostrum and inner jaw. The tip of the lower beak's rostrum in *S. inermis* is less curved and blunter, the coloration on the lateral wall of *S. japonica* appears darker (black-brown) compared to that of *S. inermis* at the maturity stage. The beaks of *S. inermis* and *S. japonica* exhibit more similarities than those of *Sepiella mangkangunga* sp. and *Sepiella weberi* within the same genus [28]. For example, in the identification and differentiation of *S. officinalis*, *S. orbignyana*, and *S. elegans*, it has been observed that the upper beak in the adult stage can be identified more accurately than the lower beak [29]. In the case of *S. inermis*, the radula consists of seven rows, including the rachidian tooth, the 1st lateral tooth, the 2nd lateral tooth, and the marginal tooth. The marginal tooth is characterized by a singular cusp-like form. In contrast, the study of [30] indicates a marginal plate; however, no marginal plate was identified in this study, and bears a resemblance to *S. japonica* [31]. This finding is consistent with the results reported by Zheng et al. [32], indicating the disappearance of the marginal plate in certain species of Sepioidea. Thus, the radula can serve as an identifying feature at the species level [33].

### 4.2. The Subcutaneous Gland's Structure

In this study, an examination of both internal and external structures was conducted from the embryonic to the adult stages. External structures in the embryonic stages were observed from stage 19 to hatching. It initiates as a small knob without color and evolves into two lobes in stages 20–22. Subsequently, in stages 23–24, it further develops into four lines, eventually culminating in the complete formation of four lobes in stage 25. Interestingly, this fully formed structure is capable of producing melanin pigment, which is responsible for the release of ink. Interestingly, the color of the gland undergoes significant changes throughout the developmental stages. In stage 19, it appears transparent, then transitions to faintly orange, orange, and dark orange or red in the hatching stage. Finally, in adults, it takes on a dark brown color [5,34–36]. In response to sudden adjustments in salinity or stressful conditions, the gland releases a coffee-brown colored fluid [37,38]. Notably, the subcutaneous gland in stage 19 has developed an external structure, while the formation of the ink sac structure begins in stage 22. From these observations, it can be concluded that there is no connection between the ink sac and the subcutaneous gland, nor does the ink transfer to the gland. The histological study of the subcutaneous gland in both the hatching and adult stages confirmed these results. Both stages exhibited four layers of membranes, including the external muscular layer (epidermis), connective tissue, muscular fibers, and inner wall membrane. The adult stage displayed thicker layers of membranes compared to the hatching stage. These trabeculae likely played a role in producing melanin glandules that connected from the basal region of the gland. The four-line trabeculae pattern in the paralarvae stage has gradually decreased to disappear in the adult stage following their growth and development. This distribution of melanin granules within the trabeculae connective tissue facilitated ink production [39]. Melanin granules, integral to the production of ink in *S. inermis*, are synthesized and composed of various cellular components including cell membranes, melanosomes, vesicles, mitochondria, Golgi apparatus, endoplasmic reticulum, and nuclei. These components work in concert within the ink sac cavity, where the melanin granules are secreted and stored, awaiting the opportune moment for expulsion [40]. However, it is essential to note that the mechanism behind the release of ink from the ink sac is stimulated by the nervous system, as reported by Derby et al. [4]. Further research is necessary to study the subcutaneous gland and its connection to the nervous system. In general, cephalopods have an ink sac [41]. The

comparison of the two glands reveals both differences and similarities in terms of structure. Previous studies on the ink sac wall in *Sepioteuthis sepioidea* and *Sepia* have reported a four-layered structure with a semicircular shape [42,43], which is similar to the structure observed in the subcutaneous gland. Understanding the structure and functioning of both the ink sac and the subcutaneous gland is crucial to comprehending the process of ink production in cephalopods in the future.

*4.3. Morphological Variations, Population Analysis, and Length–Weight Relationship*

In this study, we employed the dorsal mantle length (DML) as a metric to evaluate the accuracy of growth rates and morphometric proportions. Tables A1 and A2 present the data on various body proportions relative to DML, allowing for an analysis of different morphological characteristics. External morphometric traits of various body parts are influenced by DML [44]. Proportional measurements play a crucial role in understanding growth patterns and reflecting morphological development, especially when comparing males and females. In general, the study indicates that females have larger body sizes than males in various aspects, such as body weight (BW), total length (TL), dorsal mantle length (DML), fin length (FL), fin width (FW), etc. Interestingly, the length of the left and right arms (1–4) is slightly longer in males compared to females. This observation aligns with the findings reported by Mahadwala et al. [19], indicating that the dorsal mantle length was 35.9–93.3 mm DML for males and 29.3–98 mm DML for females at sexual maturity. Furthermore, according to the studies of Sundaram et al. [44], the length of females was 55 mm, while that of males was 48 mm. The range of data in this study coincides with that in the study conducted by [45], which found that males had a lower body weight and DML than females. The analysis revealed that females exhibited greater length and weight compared to males, with a statistically significant difference ($p < 0.05$). Additionally, the *b* value, representing the scaling relationship between body weight and dorsal mantle length, differed between females ($b = 2.3837$) and males ($b = 2.1746$). When both sexes were combined, the overall *b* value was calculated as 2.3616. These findings indicate that both males and females exhibited allometric growth. These observations are consistent with previous studies [1,19,46–48], further validating the presence of allometric growth in both males and females.

Based on the discriminant function analysis, the study aears were grouped into three distinct clusters: Group 1 consisted of the T, SK, and SNI stations, while Group 2 comprised the PT, SSM, and RN stations, showing some overlapping territories. This implies that there is relatively less variation in morphological traits within this geographical group. However, the CB station stood out from the other groups in the scatter plot, displaying clear separation without any significant overlap with the other groups (refer to Figures 10b and 11). These findings were consistent with the body weight data presented in Tables 2 and A2, providing strong evidence for the close relationship between morphological differences and the habitat within the CB station. Besides, significant differences ($p < 0.05$) were observed between males and females based on morphological variations. For instance, in Group 1, both males and females show close relationships within PT, SSM, and SNI, but there are variations in CB, SK, and T, with similar relationships observed in RN stations. The relationship between the SNI and RN stations indicates similar environmental conditions that may contribute to the observed morphological similarities within a group. The analysis of length–weight relationships (LWRs) and relative growth patterns indicated that six stations, namely T, SK, PT, SSM, RN, and CB, exhibited allometric growth. However, the SNI station demonstrated isometric growth ($b = 3.0272$). Remarkably, these results align with the highest percentage of discrimination accuracy observed for both sexes. This suggests that this specific population group is significantly influenced by conditions conducive to abundant growth and optimal environmental factors, as shown in Tables 2, 3 and A2, and Figure 12.

For instance, previous studies on four species of cuttlefish, including *Sepia officinalis*, *S. elegans*, and *S. orbignyana* distributed in the Mediterranean Sea, and *S. pharaonic* with an isometric growth rate in the eastern Mediterranean Sea, indicate that various factors,

such as temperature, food availability, or sampling methodology, might contribute to these differences [49]. The observed variations in proportions between males and females in this particular station may have influenced the results of the analysis. Consequently, the information obtained from this study may not be entirely conclusive in determining whether the localities surveyed are abundant and suitable for marine animals. To gain a more comprehensive understanding, it is necessary to collect a random sample size comprising both males and females across a wider range of sizes and developmental stages while also encompassing a broader range of localities. It is important to consider that the coastal area of Thailand is influenced by a multitude of factors, for instance, freshwater sources from land, ocean currents, temperature fluctuations, and overfishing. Previous studies on squid and cuttlefish's growth and maturity stages vary depending on different geographic regions. [5,19,50–53]. To gain a comprehensive understanding of cephalopod distribution patterns and ecological dynamics scale, it is imperative to gather genetic material data and detect morphological differences [54]. Considering the external morphological characteristics, morphological variations, and histological structure of the subcutaneous gland in *S. inermis*, this study offers valuable insights and provides a deeper understanding of geographic variations; particularly significant is the first detailed description of the subcutaneous gland's structure, which serves as a foundational framework for future in-depth investigations on this subject.

## 5. Conclusions

This study provides a comprehensive description of the external morphological diversity of *S. inermis*. In general, the study indicates that females exhibit larger body sizes than males in various aspects, including body weight (BW), total length (TL), dorsal mantle length (DML), fin length (FL), fin width (FW), etc. Female cuttlebones are broader and more pronouncedly dented than those of males, and males typically exhibit 17–19 white dots. Moreover, the examination of the external structure of the subcutaneous gland indicates its initiation in the embryonic stage at stage 19 and developmental stages to the adult stage. The ink sac and the subcutaneous gland are not connected, and there is no ink transfer to the gland. Morphometric analysis of the DML revealed significant differences ($p < 0.05$) between both sexes in different regions. The discriminant analysis also revealed significant differences ($p < 0.05$) in station SNI. Our findings indicate that both males and females display allometric growth.

**Author Contributions:** X.Z. and S.P. designed the study; S.P. wrote the original draft; X.Z., C.S. and R.X. reviewed and edited the paper; S.P. collected the samples; X.Z., S.P. and C.S. performed the experiments and analyses. All authors have read and agreed to the published version of the manuscript.

**Funding:** This work was supported by the Chinese Government Scholarship (CSC scholarship) and the support of the National Natural Science Foundation of China under Grant 32170536.

**Institutional Review Board Statement:** All experiments involving the handling and treatment of animals were conducted in strict accordance with the recommendations in the Guide for Animal Care.

**Data Availability Statement:** The raw data supporting the conclusions of this article will be made available by the authors on request.

**Acknowledgments:** The authors would like to sincerely express their gratitude to the Department of Marine Science, Faculty of Fisheries, Kasetsart University, for generously allowing the utilization of their equipment and laboratories throughout this study. Additionally, we extend our heartfelt appreciation to the Ocean University of China and the Chinese Government Scholarship (CSC scholarship) for their generous support. This research was made possible through the support of the National Natural Science Foundation of China under Grant 31672257. We would also like to convey our special appreciation to the students in China and Thailand for their invaluable assistance and contribution to the collection of specimens, which greatly facilitated the success of this investigation.

**Conflicts of Interest:** The authors declare no conflict of interest.

## Appendix A

**Table A1.** External morphological characteristics of the spineless cuttlefish, *S. inermis*, in the waters of Thailand, are described in detail.

| Morphological Index | Gentle | Number of Samples | (mm) Mean ± SD | Rage Min | Max | Percentage of Length |
|---|---|---|---|---|---|---|
| BW | Female | 193 | 46.18 ± 20.76 | 8.74 | 133.00 | |
| | Male | 203 | 32.89 ± 12.87 | 11.00 | 82.00 | |
| | Total | 396 | 39.37 ± 18.40 | 8.74 | 133.00 | |
| DML | Female | 193 | 61.34 ± 11.35 | 32.00 | 92.00 | |
| | Male | 203 | 56.91 ± 9.16 | 35.00 | 79.00 | |
| | Total | 396 | 59.07 ± 10.51 | 32.00 | 92.00 | |
| TL | Female | 193 | 111.11 ± 22.46 | 58.00 | 166.00 | 55.21% of TL |
| | Male | 203 | 104.71 ± 19.79 | 64.00 | 154.00 | 54.35% of TL |
| | Total | 396 | 107.83 ± 21.35 | 58.00 | 166.00 | 54.78% of TL |
| DMW | Female | 193 | 47.94 ± 8.00 | 24.27 | 73.82 | 78.15% of DML |
| | Male | 203 | 43.49 ± 6.36 | 23.41 | 60.16 | 76.42% of DML |
| | Total | 396 | 45.66 ± 7.53 | 23.41 | 73.82 | 77.29% of DML |
| CL | Female | 193 | 58.77 ± 11.17 | 30.10 | 85.00 | 95.81% of DML |
| | Male | 203 | 53.00 ± 8.36 | 28.00 | 76.00 | 93.14% of DML |
| | Total | 396 | 55.82 ± 10.23 | 28.00 | 85.00 | 94.49% of DML |
| CW | Female | 193 | 25.10 ± 4.99 | 12.00 | 38.00 | 40.93% of DML |
| | Male | 203 | 21.25 ± 3.18 | 14.00 | 33.00 | 37.33% of DML |
| | Total | 396 | 23.13 ± 4.58 | 12.00 | 38.00 | 39.15% of DML |
| CWt | Female | 193 | 1.57 ± 0.76 | 0.27 | 4.81 | 3.40% of BW |
| | Male | 203 | 1.13 ± 0.60 | 0.18 | 3.87 | 3.45% of BW |
| | Total | 396 | 1.35 ± 0.72 | 0.18 | 4.81 | 3.42% of BW |
| HL | Female | 193 | 30.31 ± 5.03 | 17.35 | 43.17 | 49.41% of DML |
| | Male | 203 | 28.87 ± 6.64 | 19.36 | 83.74 | 50.73% of DML |
| | Total | 396 | 19.80 ± 4.37 | 7.78 | 32.91 | 66.95% of DML |

| Morphological Index | Gentle | Number of Samples | (mm) Mean ± SD | Rage Min | Percentage of Length Max | Morphological Index |
|---|---|---|---|---|---|---|
| HW | Female | 193 | 20.62 ± 4.50 | 11.95 | 32.91 | 68.05% of HL |
| | Male | 203 | 19.01 ± 4.09 | 7.78 | 29.84 | 65.85% of HL |
| | Total | 396 | 29.57 ± 5.95 | 17.35 | 83.74 | 50.05% of HL |
| ED | Female | 193 | 8.83 ± 1.92 | 4.63 | 16.80 | 29.15% of HL |
| | Male | 203 | 8.68 ± 1.85 | 4.77 | 19.09 | 30.05% of HL |
| | Total | 396 | 8.75 ± 1.88 | 4.63 | 19.09 | 29.59% of HL |
| FL | Female | 193 | 32.36 ± 6.04 | 11.38 | 47.30 | 52.75% of DML |
| | Male | 203 | 29.29 ± 5.25 | 10.54 | 45.19 | 51.46% of DML |
| | Total | 396 | 19.80 ± 4.37 | 7.78 | 32.91 | 66.95% of DML |
| FW | Female | 193 | 11.21 ± 2.63 | 5.41 | 19.97 | 18.28% of DML |
| | Male | 203 | 10.10 ± 2.19 | 5.34 | 15.67 | 17.74% of DML |
| | Total | 396 | 10.64 ± 2.48 | 5.34 | 19.97 | 18.01% of DML |
| LI | Female | 193 | 23.01 ± 6.93 | 10.01 | 40.68 | 37.50% of DML |
| | Male | 203 | 25.47 ± 8.37 | 10.80 | 49.13 | 44.76% of DML |
| | Total | 396 | 24.27 ± 7.79 | 10.01 | 49.13 | 41.08% of DML |
| LII | Female | 193 | 23.22 ± 6.89 | 10.33 | 42.21 | 37.85% of DML |
| | Male | 203 | 24.73 ± 8.05 | 11.67 | 50.97 | 43.46% of DML |
| | Total | 396 | 24.00 ± 7.54 | 10.33 | 50.97 | 40.62% of DML |
| LIII | Female | 193 | 24.59 ± 6.83 | 11.40 | 41.73 | 40.09% of DML |
| | Male | 203 | 27.33 ± 7.41 | 12.09 | 49.24 | 48.02% of DML |
| | Total | 396 | 25.99 ± 7.25 | 11.40 | 49.24 | 44.00% of DML |
| LIV | Female | 193 | 32.64 ± 8.26 | 15.33 | 59.91 | 53.20% of DML |
| | Male | 203 | 32.82 ± 9.06 | 16.69 | 58.46 | 57.66% of DML |
| | Total | 396 | 32.73 ± 8.67 | 15.33 | 59.91 | 55.40% of DML |
| RI | Female | 193 | 23.07 ± 7.26 | 9.93 | 39.35 | 37.61% of DML |
| | Male | 203 | 25.51 ± 8.39 | 11.10 | 50.64 | 44.82% of DML |
| | Total | 396 | 24.32 ± 7.95 | 9.93 | 50.64 | 41.17% of DML |
| RII | Female | 193 | 23.51 ± 7.13 | 10.27 | 41.72 | 38.33% of DML |
| | Male | 203 | 25.27 ± 8.19 | 11.15 | 48.10 | 44.40% of DML |
| | Total | 396 | 24.41 ± 7.73 | 10.27 | 48.10 | 41.32% of DML |
| RIII | Female | 193 | 24.86 ± 7.16 | 12.21 | 42.54 | 40.53% of DML |
| | Male | 203 | 27.44 ± 8.01 | 12.37 | 49.19 | 48.22% of DML |
| | Total | 396 | 26.18 ± 7.71 | 12.21 | 49.19 | 44.32% of DML |
| RIV | Female | 193 | 31.98 ± 8.96 | 16.82 | 64.52 | 52.14% of DML |
| | Male | 203 | 32.67 ± 9.06 | 16.63 | 56.61 | 57.41% of DML |
| | Total | 396 | 32.34 ± 9.01 | 16.63 | 64.52 | 54.74% of DML |

**Table A2.** Detailed analyses of external morphological characteristics of spineless cuttlefish, *S. inermis* were conducted at each station in the waters of Thailand.

| | | | | Station | | | |
|---|---|---|---|---|---|---|---|
| Location | TT | CB | SSM | PT | SNI | SK | RN |
| Number of Samples | 30 | 61 | 80 | 75 | 30 | 45 | 75 |
| **Morphological Index (mm mean ± SD)** | | | | | | | |
| BW | 58.7 ± 21.93 | 21.61 ± 6.07 | 41.05 ± 14.91 | 39.49 ± 14.07 | 47.43 ± 20.47 | 54.93 ± 18.14 | 31.60 ± 12.00 |
| TL | 133.53 ± 17.08 | 80.48 ± 9.37 | 111.76 ± 9.38 | 112.12 ± 11.61 | 122.5 ± 15.69 | 124.2 ± 16.30 | 95.62 ± 13.37 |
| DML | 69.53 ± 10.16 | 48.07 ± 6.22 | 61.3 ± 9.10 | 62.43 ± 7.95 | 61.17 ± 7.87 | 68.02 ± 7.15 | 51.89 ± 6.89 |
| DMW | 54.02 ± 6.39 | 36.99 ± 4.57 | 46.51 ± 5.17 | 44.43 ± 4.30 | 52.81 ± 7.05 | 52.17 ± 5.66 | 42.91 ± 6.04 |
| CL | 65.07 ± 10.11 | 43.35 ± 5.10 | 57.38 ± 9.02 | 59.69 ± 8.64 | 56.17 ± 7.82 | 64.38 ± 6.55 | 51.43 ± 6.46 |
| CW | 27.00 ± 5.41 | 18.44 ± 2.56 | 24.11 ± 4.37 | 23.92 ± 3.97 | 24.17 ± 3.70 | 26.24 ± 3.32 | 21.25 ± 3.39 |
| CWt | 2.16 ± 1.07 | 0.68 ± 0.22 | 1.24 ± 0.46 | 1.40 ± 0.62 | 1.69 ± 0.68 | 2.07 ± 0.52 | 1.04 ± 0.45 |
| HL | 33.48 ± 4.02 | 24.21 ± 2.95 | 28.62 ± 4.19 | 31.53 ± 3.67 | 30.80 ± 3.61 | 36.46 ± 9.84 | 26.80 ± 2.93 |
| HW | 24.19 ± 5.05 | 15.27 ± 2.07 | 20.57 ± 3.26 | 19.74 ± 3.80 | 20.03 ± 2.50 | 24.78 ± 3.00 | 17.88 ± 3.23 |
| ED | 11.06 ± 2.62 | 6.33 ± 0.72 | 8.53 ± 1.47 | 9.52 ± 0.96 | 10.16 ± 1.25 | 9.98 ± 1.53 | 7.97 ± 0.79 |
| FL | 37.28 ± 4.46 | 23.42 ± 3.03 | 31.30 ± 4.82 | 31.52 ± 3.75 | 34.12 ± 4.96 | 35.97 ± 3.28 | 28.43 ± 4.84 |
| FW | 14.09 ± 1.85 | 7.72 ± 1.32 | 10.98 ± 2.05 | 1.48 ± 7.42 | 12.16 ± 2.48 | 12.25 ± 1.87 | 9.38 ± 1.64 |
| LI | 30.33 ± 4.54 | 15.58 ± 2.65 | 24.83 ± 6.89 | 26.01 ± 4.95 | 29.20 ± 5.32 | 32.71 ± 9.20 | 19.54 ± 4.13 |
| LII | 30.96 ± 4.52 | 16.00 ± 2.54 | 24.80 ± 6.69 | 24.36 ± 4.75 | 27.80 ± 6.08 | 32.89 ± 8.38 | 19.63 ± 4.12 |
| LIII | 32.64 ± 5.08 | 18.01 ± 3.15 | 26.40 ± 6.29 | 27.08 ± 4.00 | 30.11 ± 6.88 | 33.72 ± 7.83 | 22.03 ± 4.28 |
| LIV | 46.74 ± 7.72 | 22.88 ± 4.06 | 33.05 ± 7.23 | 33.19 ± 4.32 | 36.90 ± 5.28 | 39.72 ± 8.29 | 28.48 ± 4.84 |
| RI | 30.39 ± 4.96 | 15.39 ± 2.74 | 25.13 ± 7.12 | 26.08 ± 5.28 | 29.23 ± 5.07 | 32.91 ± 9.09 | 19.42 ± 4.06 |
| RII | 30.95 ± 6.00 | 15.60 ± 2.75 | 25.38 ± 6.79 | 25.72 ± 5.03 | 28.74 ± 5.12 | 32.69 ± 8.57 | 19.93 ± 3.74 |
| RIII | 33.53 ± 5.54 | 17.38 ± 2.70 | 26.77 ± 6.91 | 27.52 ± 4.74 | 31.22 ± 5.92 | 34.13 ± 8.17 | 21.66 ± 3.88 |
| RIV | 45.87 ± 10.31 | 22.74 ± 3.76 | 32.57 ± 7.73 | 33.67 ± 4.65 | 35.65 ± 6.55 | 38.81 ± 9.17 | 27.93 ± 5.11 |

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
