# Peer review of "Morphological Variation and New Description of the Subcutaneous Gland of Sepiella inermis (Van Hasselt, 1835) in Thai Waters"

_diversity, doi:10.3390/d16030138_

Round 1
Reviewer 1 Report
Comments and Suggestions for Authors
At the moment your paper cannot be published - it is suffering from many errors and poor presentation. Details on attached file.

Comments on the Quality of English LanguageMain problem is syntax and poor understanding of English terminology. Some sentences have no real meaning because wrong words were used.
Reviewer 2 Report
Comments and Suggestions for Authors
Dear Authors
This manuscript is a great contribution to knowing more details and better presentation of the species under study.
In my opinion, the methods and results need to be better explained for a better understanding.
Roper and Voss 1983 have not been cited in the methodology on morphometric measurements.
Details of the cluster analysis (method: upgma?, distance matrix: Euclidean?) are missing.
In the descriptions of the peaks and radula, there is no mention of the size of the structures or measurements made. In Tables 2 and 3 explain the abbreviated names of the localities in the legends. The same for cluster fig 13 explains the abbreviations and the distance used.
In results and discussion explain that in L-W b<3 represents allometric growth.
I hope to see a better version of the manuscript soon.
All the best.
Reviewer 3 Report
Comments and Suggestions for Authors
This is an excellent contribution requiring a comprehensive revision as set out below. The authors make frequent mentions of the importance of this work and the need for further research but fail to identify the target audience, They mention there work specifically, then talk about cephalopod research in general that I find confusing.
Comments on the Quality of English LanguageThis contribution is spoiled by being overly wordy and at times has poor sentence structure.
For example, line 16, "The external morphology and variations of Sepiella inermis elicit great fascination". Should at least say to who, and "great fascination"? I would say interest or promise.
Line 17, "However, there remains a significant lack of information, particularly the histology of the subcutaneous gland". I think this should read, "However, there is a lack of information regarding the histology of the subcutaneous gland". Then "An external morphological analysis etc.
Line 30, "indicated that distinct growth rates ARE influenced by allometry.
Round 2
Reviewer 1 Report
Comments and Suggestions for Authors
You have ignored a serious consideration to some issues raised in a review. For example, growth, sample vs. population, possibility of artifacts during preparation of specimens (SEM, paraffin intrusions in histological slides). Your attention to detail should be improved. However, your MS has improved.

Comments on the Quality of English LanguageStill serious problems. Some sentences not finished, meaning not clear.
Round 3
Reviewer 1 Report
Comments and Suggestions for Authors
Still a number of issues not attended to. See attached file.

Comments on the Quality of English LanguageYou MUST run your MS via native English speaker. Otherwise many of your important sentences will create confusion. If not attended to, your MS may be rejected by the Editors.
